# Parallel computing efficiency of SWAN 40.91

Christo Rautenbach[1, 2, 3, 4], Julia C. Mullarney[4], Karin R. Bryan[4]

[1] Institute for Coastal and Marine Research, Nelson Mandela University, South Africa
[2] Department of Oceanography and Marine Research Institute, University of Cape Town, South Africa
[3] Research and development, MetOcean (a division of the Metrological Service), Raglan, New Zealand
[4] Environmental Research Institute, University of Waikato, Hamilton, New Zealand

*Correspondence to*: Christo Rautenbach (rautenbachchristo@gmail.com)

**Abstract.** Effective and accurate ocean and coastal wave predictions are necessary for engineering, safety, and recreational purposes. Refining predictive capabilities is increasingly critical to reduce the uncertainties faced with a changing global wave
climatology. Simulating WAves in the Nearshore (SWAN) is a widely used spectral wave modelling tool employed by coastal engineers and scientists, including for operational wave forecasting purposes. Fore- and hindcasts can span hours to decades and a detailed understanding of the computational efficiencies is required to design optimized operational protocols and hindcast scenarios. To date, there exists limited knowledge on the relationship between the size of a SWAN computational domain and the optimal amount of parallel computational threads/ cores required to execute a simulation effectively. To test
the scalability, a hindcast cluster of 28 computational threads/ cores (1 node) was used to determine the computation efficiencies of a SWAN model configuration for southern Africa. The model extent and resolution emulate the current operational wave forecasting configuration developed by the South African Weather Service (SAWS). We implemented and compared both OpenMP and the Message Passing Interface (MPI) distributing memory architectures. Three sequential simulations (corresponding to typical grid cell numbers) were compared to various permutations of parallel computations using
the speed-up ratio, time saving ratio and efficiency tests. Generally, a computational node configuration of 6 threads/ cores produced the most effective computational set-up based on wave hindcasts of one-week duration. The use of more than 20 threads/ cores resulted in a decrease in speed-up ratio for the smallest computation domain, owing to the increased sub-domain communication times for limited domain sizes.

*Keywords: SWAN, Parallel computing, Forecasting, Hindcasting, South Africa*

## 1 Introduction

The computational efficiency of Met-ocean (Metrological-Ocean) modelling has been the topic of ongoing deliberation for decades. The applications range from long-term atmospheric and ocean hindcast simulations to the fast responding simulations related to operational forecasting. Long-duration simulations are usually associated with climate change related research, with simulation periods of at least 30-years across multiple spatial and temporal resolutions needed to capture key oscillations
(Babatunde et al., 2013). Such hindcasts are frequently used by coastal and offshore engineering consultancies for purposes

such as those related to infrastructure design (Kamphuis, 2020), or environmental impact assessments (Frihy, 2001; Liu et al., 2013).

Operational (or forecasting) agencies are usually concerned with achieving simulation speeds that would allow them to accurately forewarn their stakeholders of immediate, imminent and upcoming met-ocean hazards. The main stakeholders are usually other governmental agencies (e.g. disaster response or environmental affairs departments), commercial entities and the public. Both atmospheric and marine forecasts share similar numerical schemes that solve the governing equations and thus share a similar need in computational efficiency. Fast simulation times are also required for other forecasting fields such as hydrological dam-break models (e.g. Zhang, et al., (2014)). Significant advancement in operational forecasting can be made by examining the way in which the code interfaces with the computation nodes, and how results are stored during simulation.

Numerous operational agencies (both private and public) makes use of Simulating Waves in the Nearshore (SWAN) to predict nearshore wave dynamics (refer to Genseberger & Donners, (2020) for details regarding the SWAN numerical code and solution schemes). These agencies include the South African Weather Service (e.g. Rautenbach, et al., (2020)), MetOcean Solutions (a division of the Metrological Office of New Zealand) (e.g. de Souza, et al., (2020)), the United Kingdom MetOffice (e.g. O'Neill et al., (2016)) and the Norwegian Metrological Service (e.g. Jeuring, et al., (2019). In general, these agencies have substantial computational facilities but nonetheless still face the challenge of optimizing the use of their computational clusters between various models (being executed simultaneously). These models may include atmospheric models (e.g. the Weather Research and Forecasting (WRF) model), Hydrodynamic models (e.g. Regional Ocean Modeling System (ROMS) and the Semi-implicit Cross-scale Hydroscience Integrated System Model (SCHISM)) and spectral waves models (e.g. Wave Watch III (WW3) and SWAN (Holthuijsen, 2007; The SWAN team, 2019b; The SWAN team, 2019)). Holthuijsen, (2007) presents a theoretical background to the spectral wave equations, wave measurement techniques and statistics as well as a concluding chapter the theoretical analysis to the SWAN numerical model. There must also be a balance between hindcast and forecast priorities and client needs. Some of these agencies use a regular grid (instead of irregular grids (e.g. Zhang, et al., (2016)), with nested domains in many of their operational and hindcast projects. Here, we focus only on the computational performance of a structured regular grid (typically implemented for spectral wave models).

Kerr et al., (2013) performed an inter-model comparison of computational efficiencies by comparing SWAN, coupled with ADCIRC, and the NOAA official storm surge forecasting model, Sea, Lake, and Overland Surges from Hurricanes (SLOSH); however, did not investigate the optimal thread usage of a single model. Other examples of a coupled wave and storm surge model computational benchmarking experiments include Tanaka, et al, (2011) and Dietrich et al., (2012) who used a unstructured meshes to simulate waves during Hurricanes Katrina, Rita, Gustav and Ike in the Gulf of Mexico. Results from these models were presented on a log-log scale and their experimental design tested computational thread numbers not easily obtainable by smaller agencies and companies. The latter rather require sequential versus paralleled computational efficiencies using smaller scale efficiency metrics. Genseberger & Donners, (2015), explored the scalability of SWAN using a case study focused on the Wadden Sea in the Netherlands. By investigating the efficiency of both the OpenMP (OMP) and MPI version of the then current SWAN, they found that the OpenMP was more efficient on a single node. They also proposed a hybrid

version of SWAN, to combine the strengths of both implementations of SWAN: using OpenMP to more optimally share memory and MPI to distribute memory over the computational nodes.

Here we build on the case study of Genseberger & Donners using results produced in the present study for southern Africa, to answer the following research questions: 1) when using SWAN, is it always better to have as many threads/ cores as possible

available to solve the problem at hand? 2) What is the speed-up relationship between number of threads/ cores and computational grid size? 3) At what point (number of threads/ cores) does the domain sub-communications start to make the whole computation less effective? 4) What is the scalability of a rectangular grid, SWAN set-up?

**Methodology and background**

Details of the model configuration can be found in Rautenbach, et al., (2020a) and Rautenbach, et al., (2020b). The

computational domain (refer to Figure 1) and physics used here were the same as presented in those studies.

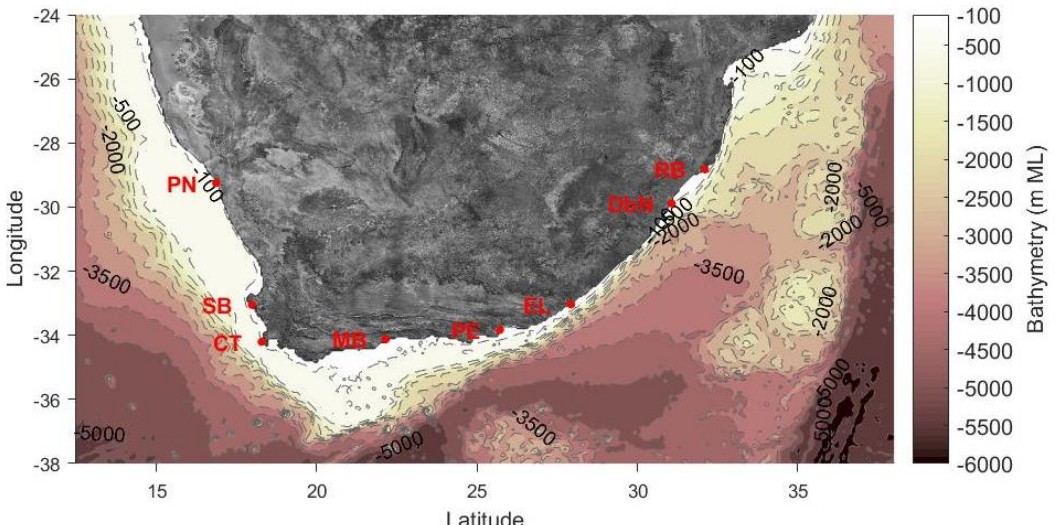

**Figure 1: SWAN model extent and associated bathymetry. The location of all the major coastal towns are also provided via acronyms as follows: Port Nolloth (PN), Saldanha Bay (SB), Cape Town (CT), Mossel Bay (MB), Port Elizabeth (PE), East London (EL), Durban (DN) and Richards Bay (RB).**

All computations were performed on Intel Xeon Gold E5-2670, 2.3GHz computational nodes. Twenty-eight threads/ cores each with 96 GB RAM were used with1 Gbyte/s inter-thread communication speed. Given that the present study was performed using a single computational node, inter-node communication speeds are not considered. Thus, given a computational node with similar processing speed, the present study should be reproducible. In general, these node specifications are reasonably

standard and therefore the present study is an acceptable representation of the SWAN scalability parameters.

SWAN 40.91 was implemented with the Van der Westhuysen whitecapping formulation (van der Westhuysen, et al., 2007) and Collins bottom friction correlation (Collins, 1972) with a coefficient value of 0.015. Fully spectral wave boundary conditions were extracted from a global Wave Watch III model at 0.5 geographical degree resolution.

Here, the main aim was not the validation of the model but rather to quantify the relative computational scalabilities, as described at the end of the previous section. However, it should be noted that no nested domains were employed during the present study. Only the parent domain was used as a measure for scalability. The computational extent given in Rautenbach, Barnes, et al., (2020) (a) and Rautenbach, et al., (2020) (b) contains numerous non-wet grid cells that are not included in the computational expense of the current study. In Table 1, the size of the computational domain and resolution, together with the labelling convention are given. For clarity, we define the resolutions as low, medium and high, denoted L, M and H, respectively, in the present study (noting that given the domain size, these resolutions would be classified as intermediate to high regional resolution for operational purposes).

**Table 1: SWAN grid resolution, grid cell numbers and reference labels.**

| Label | SWAN grid resolution | Computational grid cell number |
|-------|---------------------|-------------------------------|
| L | 0.1000 | 31 500 |
| M | 0.0625 | 91 392 |
| H | 0.0500 | 142 800 |

The test for scalability ability of a model used here was the ability to respond to an increased number of computations with an increasing amount of resources. In the present study these resources are computational threads/ cores. An arbitrary week of computations were performed to assess model performance. Model spin-up was done via a single stationary computation. The rest of the computation was performed using a non-stationary computation using an hourly time-step, which implied wind-wave generation within the model occurred on the timescale of the wind forcing resolution. The grid resolutions used in the present study corresponded to 0.1, 0.0625 and 0.05 geographical degrees. Local bathymetric features were typically resolved through downscaled, rotated, rectangular grids, following the methodology employed by Rautenbach, et al., (2020) (a). A nested resolution increase of more than 5-times is also not recommended (given that the regional model is nested in the global Wave Watch III output at 0.5 geographical degree resolution, (Rautenbach, et al., 2020) (a). Given these constraints, these resolutions represent realistic and typical SWAN model set-up, for both operational and hindcast scenarios.

The three main metrics for estimating computational efficiency are: the *Speed-up*, *Time saving* and *Efficiency ratios*. A fourth parameter, and arguably the most important, is the *Scalability* and is estimated using the other three parameters as metrics. The *Speed-up* ratio is given as:

$$S_p = T_1/T_p \tag{1}$$

where $T_1$ is the time in seconds it takes for a sequential computation on one thread and $T_p$ is the time a simulation takes with $p$
computational threads/ cores (Zhang et al., 2014).

The *Time saving* ratio is given by:

$$T_1 S_p = (T_1 - T_p)/T_1 \tag{2}$$

and the *Efficiency* ratio is defined as:

$$E_p = S_p/p. \tag{3}$$

The Scalability of SWAN was tested based on the Speed-up ratios for the grid resolutions in Table 1.

Zafari, Larsson, & Tillenius, (2019) recently presented some of the first results investigating the effect of different compilers on the scalability of a shallow water equation solver. Their experiments compared a model compiled with GNU Compiler Collection (gcc) 7.2.0 and linked with OpenMPI and Intel C++ compilers with Intel MPI for relatively small computational problems. Their numerical computation considered models with 600K, 300K and 150K grid cell sizes (what they called matrix size). These computational grid sizes were deemed "small", but they still acknowledged the significant computational
resources required to execute geographical models of this size due to the large number of time steps undertaken to solve these problems.

From a practical point of view, regular SWAN grids will rarely be used in dimensions exceeding the resolutions presented in the previous section. The reason for this statement is twofold: 1) to downscale a spectral wave model from a global resolution
to a regional resolution should not exceed a five-times refinement factor and 2) when reasonably high resolutions are required in the nearshore (to take complex bathymetric features into account), nested domain are preferred. The reasoning will be different for an unstructured grid approach (Dietrich et al., 2012). Given these limitations with the widely used structured SWAN grid approach, SWAN grids will almost exclusively be deemed as a low spatial computational demand model. Small tasks create a sharp drop in performance via the Intel C++ compiler due to the "work stealing" algorithm, aimed at balancing
out the computational load between threads/ cores (Zafari et al., 2019). In this scenario, the threads/ cores compete against each other resulting in an unproductive simulation. In our experiments, each task performed via Intel was approximately 13-times faster but the overall performance was 16-time slower than the equivalent gcc compiled version of the compiled shallow water model presented by Zafari et al. (2019).

**Results**

In Figure 2, the computational scalability of SWAN is given as a function of number of computational threads/ cores. Figure 2 (a) shows the computational time in seconds and here the model resolutions grouped together with not much differentiation between them. These results also highlight the need for performance metrics, like described in the previous section. From Figure 2 (b) the MPI version of SWAN is more efficient for all the computational domain sizes. There is also a clear grouping between OMP and MPI. Figure 2 (c) presents the speed-up ratios and clearly indicates that the MPI version of SWAN

outperforms the OMP version. The closer the result are to the 1:1 line, the better the scalability.

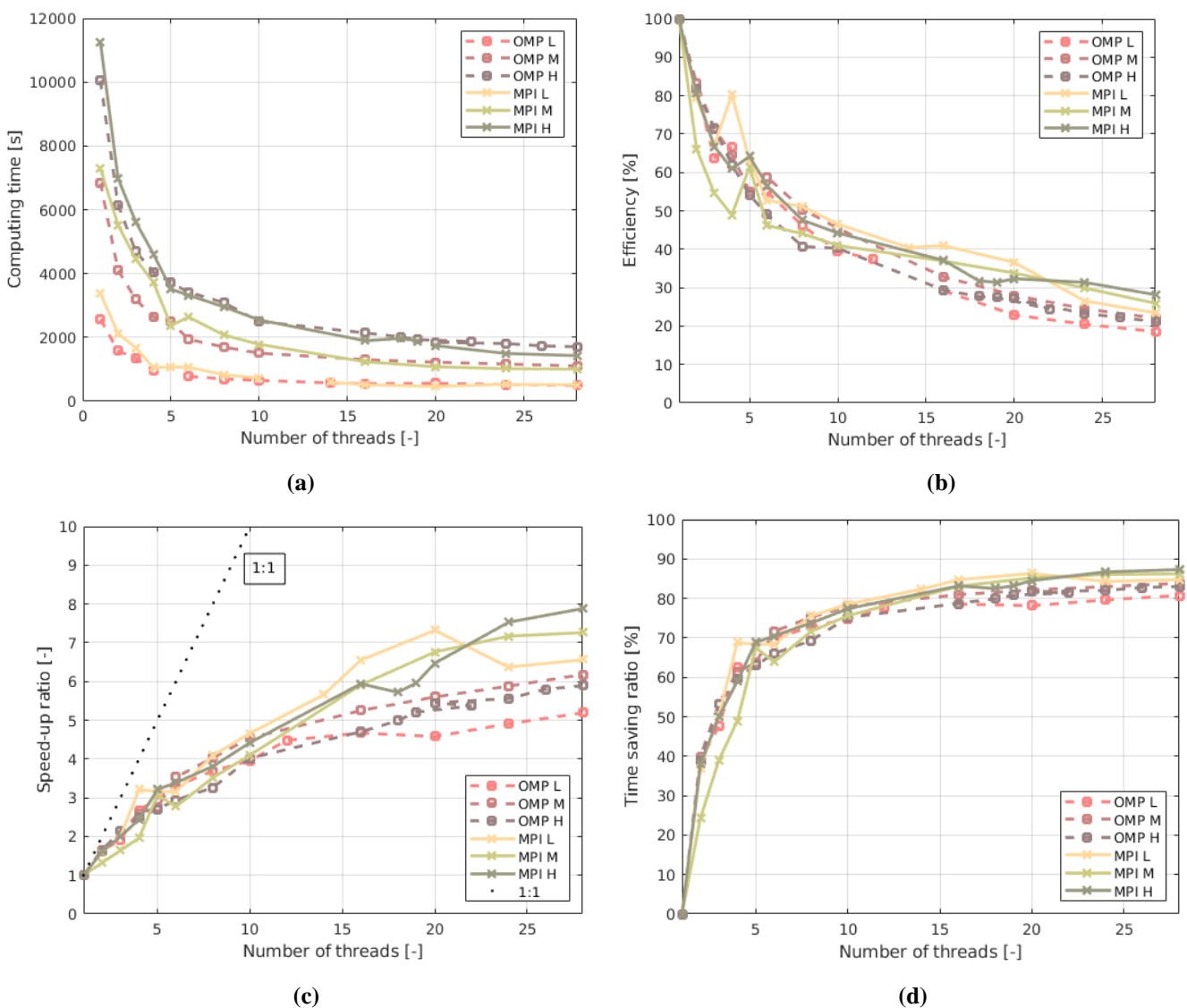

**Figure 2: Model performance as a function of the number of computational threads/ cores. (a) Computing time in seconds, (b) Efficiency (Equation (3)), (c) Speed-up ratio (Equation (1)) and (d) the Time saving ratio (Equation (2)).**

Near linear speed up is observed for a small number of computational threads/ cores. This result agrees with those reported by Zafari et al., (2019). In Figure 2 (d) the results are expressed via the time saving ratio. In this case, the curves start to asymptote with thread counts larger than approximately 6.

## Discussion

The behaviour noted in the results is similar to the dam breaking computational results reported by S. Zhang et al., (2014). Genseberger & Donners, (2020) present the latest finding on the scalability and benchmarking of SWAN. However, their focus was quantifying the performance of their new hybrid version of SWAN. In their benchmarking experiments (for the Wadden Sea, in the Netherlands), they obtained different results to Figure 2 (a), with OMP generally producing faster wall-clock computational times. They also considered the physical distances between computational threads/ cores and found that this parameter has a negligible effect compared to differences between OMP and MPI, over an increasing number of threads/ cores. Their benchmarking also differed from the results presented here as they only provided results as a function of node number. Each one of their nodes consisted of 24 threads/ cores. In the present study, the benchmarking of a single node (28 threads/ cores) is evaluated compared with a serial computation on a single thread. For benchmarking, without performance metrics, they found that the wall clock times, for the iterations and not a full simulation, reached a minimum (for large computational domains) at 16 nodes ($16 \times 24$ threads/ cores) for the MPI SWAN and 64 nodes ($64 \times 24$ threads/ cores) for the hybrid SWAN. These results were based on using the Cartesius 2690 v3 (Genseberger and Donners, 2020). With the hybrid SWAN, the optimal wall-clock time turn point, for iterations, increased with increased number of computational cells. All of the reported turn points (optimal points) occurred at node counts well above 4 nodes ($4 \times 24$ threads/ cores). The wall-clock performance estimation of Genseberger & Donners, (2015) did however indicate similar result to those presented in Figure 2 (a), with OMP running faster than MPI for small thread/ core counts. For larger thread/ core counts MPI performs better in the present study. This difference in performance is probably related to the particular hardware configuration (Genseberger and Donners, 2015). It must still be noted that with an increased number of nodes, and thus threads/ cores, the total computational time should continue to decrease up until the point where the internal domain decomposition, communication efficiencies, starts to outweigh the gaining of computational power. Based on results of Genseberger & Donners, (2020), we can estimate that, for our node configuration and region of interest, the communication inefficiencies will become dominant at approximately 16 nodes ($16 \times 24$ threads/ cores). One of the possible explanations for the non-perfect speed-up observed in Figure 2 (c) is related to the computational domain partition methods used, and the wet and dry (or active and inactive) points definitions in the model. In the present study the dry points were the bathymetry or topography values above Mean Sea Level (MSL). The employed partition method is currently stripwise because of the underlying parallel technique, namely the wavefront method

(Genseberger and Donners, 2015; Zijlema, 2005). The stripwise partition is thus potentially not the most effective method to optimize speed-up. In the present study this partition leads to an optimal point around 6 threads/ cores without losing too much parallel efficiency. In general, increasing the number of threads/ cores will still produce results faster, but in an increasingly inefficient manner. This trend is clear from Figure 2 (c) and (d) where the total computational time (speed-up ratio, time saving ratio) does not scale linearly with the increasing number of threads/ cores. The ideal scenario (linear scalability) would be if the computational results followed the 1:1 line in Figure 2 (c). In Figure 2 (d) the non-linear, flattening of the time saving ratio is also evident, although the ratio still slightly increases beyond 6 threads/ cores. This result implies the total computational time will marginally decrease with increasing number of threads/ cores. This marginal decrease in computational time could however still make significant differences in the total simulation times when extensive simulation periods are considered.

## Conclusion

The present study investigated the scalability of SWAN, a widely used spectral wave model. Three typical wave model resolutions were used for these purposes. Both the OpenMP (OMP) and the Message Passing Interface (MPI) implementations of SWAN were tested. The scalability is presented via three performance metrics: the efficiency, speed-up ratio and the timesaving ratio. The MPI version of SWAN outperformed the OMP version based on all three metrics. The MPI version of SWAN performed best with the largest computational domain resolution, resulting in the highest speed-up ratios. The time saving ratio indicated a decrease after approximately six computational threads/ cores. This result suggests that six threads/ cores are the most effective configuration for executing SWAN. The largest increases in speed-up and efficiency was observed with small thread counts. According to Genseberger & Donners, (2020), computational times decrease up to ~16 nodes (16 × 24 threads/ cores), indicating the wall-clock optimal computational time for their cases study. This result suggests that multiple nodes will be required to reach the optimal wall-clock computational time – even though this turn point might not be the most efficient computational configuration. Ultimately, the efficiencies recommended here can improve operational performance substantially, particularly when implemented over the range of modelling software needed to produce useful metocean forecasts. Future studies might consider investigating the scalability employing a gcc compiler.

## Code/Data availability

The open source version of SWAN was run for the purposes of the present study. SWAN maybe be downloaded from here: http://swanmodel.sourceforge.net/. To ensure a compatible version of SWAN remains available, the current, latest version of SWAN is permanently archive here: https://hdl.handle.net/10289/14269. The bathymetry used for the present study may be downloaded here: https://www.gebco.net/ and the wind forcing may be found here: https://climatedataguide.ucar.edu/climate-data/climate-forecast-system-reanalysis-cfsr.

## Author contribution

Dr. C. Rautenbach conceptualised the study, executed the experiments and wrote the manuscript. He also secured the publication funding. Dr. J. C. Mullarney and Professor K. R. Bryan reviewed the manuscript.

## Competing interests

No conflict of interests.

## Funding

This research was funded by the National Research Foundation of South Africa (Grant Numbers: 116359).

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
