# Peer review of "Parallel computing efficiency of SWAN 40.91"

_Geoscientific Model Development, 2020_

## Short Comment (SC1) · 21 Dec 2020

Dear authors,

in my role as Executive editor of GMD, I would like to bring to your attention our Editorial version 1.2:

https://www.geosci-model-dev.net/12/2215/2019/

This highlights some requirements of papers published in GMD, which is also available on the GMD website in the 'Manuscript Types' section: http://www.geoscientific-model-development.net/submission/manuscript_types.html

In particular, please note that for your paper, the following requirements have not been met in the Discussions paper:

[Figure]

- "The main paper must give the model name and version number (or other unique identifier) in the title."

- Code must be published on a persistent public archive with a unique identifier for the exact model version described in the paper or uploaded to the supplement, unless this is impossible for reasons beyond the control of authors. All papers must include a section, at the end of the paper, entitled "Code availability". Here, either instructions for obtaining the code, or the reasons why the code is not available should be clearly stated. It is preferred for the code to be uploaded as a supplement or to be made available at a data repository with an associated DOI (digital object identifier) for the exact model version described in the paper. Alternatively, for established models, there may be an existing means of accessing the code through a particular system. In this case, there must exist a means of permanently accessing the precise model version described in the paper. In some cases, authors may prefer to put models on their own website, or to act as a point of contact for obtaining the code. Given the impermanence of websites and email addresses, this is not encouraged, and authors should consider improving the availability with a more permanent arrangement. Making code available through personal websites or via email contact to the authors is not sufficient. After the paper is accepted the model archive should be updated to include a link to the GMD paper.

Please provide the version number of SWAN in the title of your revised manuscript.

As the websites cited in the articles code availability section are not persistent archives, please provide a persistent release for the exact source code version used for the publication in this paper. As explained in https://www.geoscientific-model-development.net/about/manuscript_types.html the preferred reference to this release is through the use of a DOI which then can be cited in the paper.

Yours, Astrid Kerkweg

---

## Referee Comment (RC1) · Anonymous Referee #1 · 4 Jan 2021

General comments

The manuscript highlights the parallel performance of the SWAN model, as a part of the operational weather forecasting in South Africa.

The manuscript is well-written and the authors set out their objectives and results very clearly. I find the results of the manuscript interesting which, in my opinion, it should be typified as a model evaluation paper rather than a development and technical paper. However, I am not sure whether the results are reproducible as they also depend on the used hardware. It might be good to highlight some additional info with respect to this aspect. (See also my comment no. 2 below.)

Specific comments

1) The notion "thread" is a bit confusing for the MPI adepts, which should be "core".
May be a combi "thread/core" would be a better wording.

2) In general, within a single node (containing a number of threads/cores) OpenMP is more efficient than MPI. So, contrary to the study of Genseberger and Donner (2015, 2020), the results of your study contradict this general statement. Do you have an explanation for this? Perhaps, you may add some technical info concerning the used hardware with respect to this aspect (memory I/O, network, etc.)

3) One of the possible reason why a perfect speed-up cannot be obtained (see Fig. 1) is the domain partition of the computational grid and also the wet/dry (or active/inactive) points. The employed partition is the stripwise one which is because of the underlying parallel technique, namely the wavefront method. See Genseberger and Donner (2015) and also Zijlema (2005). The stripwise partition might not be the most optimal one with respect to the speed-up. In this specific case, it leads to a maximum of 6 threads/cores without too much sacrificing parallel efficiency.

It would be good to highlight this aspect.

Added reference: M. Zijlema. Parallelization of a nearshore wind wave model for distributed memory architectures. In Parallel Computational Fluid Dynamics - Multidisciplinary applications, pages 207-214. Elsevier Science, 2005.

Do you have active/non-active grid points in your model schematization? Can you comment on this?

Technical corrections, etc.

line 103: ration -> ratio

line 108: compliers -> compilers

line 156: (16 x 25 threads) -> (16 x 24 threads)

line 157: (16 x 24 threads) -> (64 x 24 threads)

line 234: please change version number; also suggested to add the Technical Manual of SWAN besides the User Manual, as it contains the details of both physics and numerics
* * *

---

## Short Comment (SC2) · 11 Jan 2021

Thank you for your interest in our paper and for the discussion. Herewith inline responses to the questions, comments and corrections: SC1: Please provide the version number of SWAN in the title of your revised manuscript. As the websites cited in the articles code availability section are not persistent archives, please provide a persistent release for the exact source code version used for the publication in this paper. As explained in https://www.geoscientific-modeldevelopment.net/about/manuscript_types.html the preferred reference to this release is through the use of a DOI which then can be cited in the paper.

Author response: The version number has been added to the title. The current supported versions of SWAN are 41.20 and 41.31. From the following link we can see that the latest version of SWAN is fully compatible with version 40.91, used in the present

study (http://swanmodel.sourceforge.net/modifications/modifications.htm). The additions to the newer versions are just more choices in the parametrization models used to describe certain physical phenomena. Thus, executing the model with this latest version, together with the settings presented in this study, should produce identical results. To be sure readers of the paper can readily download all versions (Linux, Windows etc.) of this code, the current version of the software will be added to a university website associated with this paper. A link to this permanent cite will be added to the final version of the paper. We are currently in the process of establishing that online location.

---

## Referee Comment (RC2) · Anonymous Referee #1 · 12 Jan 2021

Thank you for your response. However, I can not find the newest version of the preprint.

---

## Short Comment (SC3) · 12 Jan 2021

Author response: Thank you very much for the very useful and insightful comments. It is much appreciated. Please find the inline responses below:

1) The notion "thread" is a bit confusing for the MPI adepts, which should be "core". May be a combi "thread/core" would be a better wording.

Author response: All references to "thread" have been replaced with "thread/ core". 2) In general, within a single node (containing a number of threads/cores) OpenMP is more efficient than MPI. So, contrary to the study of Genseberger and Donner (2015, 2020), the results of your study contradict this general statement. Do you have an explanation for this? Perhaps, you may add some technical info concerning the used hardware with respect to this aspect (memory I/O, network, etc.)

[Figure]

Author response: OMP performs better at small thread numbers where MPI does better at larger thread numbers. This is illustrated in Figure 1 (a) and (b). This was clarified in the text on line 160 to 165. In Genseberger they note: "So, for this hardware the OpenMP version is twice as efficient as the MPI version." It must be related to differences in the hardware. Details regarding the memory and network has been added in Methodology Section. They also looked at the Wadden Sea while this is a southern African benchmarking study. 3) One of the possible reason why a perfect speed-up cannot be obtained (see Fig. 1) is the domain partition of the computational grid and also the wet/dry (or active/inactive) points. The employed partition is the stripwise one which is because of the underlying parallel technique, namely the wavefront method. See Genseberger and Donner (2015) and also Zijlema (2005). The stripwise partition might not be the most optimal one with respect to the speed-up. In this specific case, it leads to a maximum of 6 threads/cores without too much sacrificing parallel efficiency. It would be good to highlight this aspect. Added reference: M. Zijlema. Parallelization of a nearshore wind wave model for distributed memory architectures. In Parallel Computational Fluid Dynamics - Multidisciplinary applications, pages 207-214. Elsevier Science, 2005. Do you have active/non-active grid points in your model schematization? Can you comment on this? Technical corrections, etc. line 103: ration -> ratio line 108: compliers -> compilers line 156: (16 x 25 threads) -> (16 x 24 threads) line 157: (16 x 24 threads) -> (64 x 24 threads)

Author response: Thank you for this great recommendation. These details and references have been added on lines 170 onwards, at the end of the Discussion section. Yes, the African continent computational points are inactive. This details has also been added in the Discussion section. All these corrections were made, thank you.

line 234: please change version number; also suggested to add the Technical Manual of SWAN besides the User Manual, as it contains the details of both physics and numerics Author response: Corrections made, and references added.

———————————————

---

## Short Comment (SC4) · 12 Jan 2021

I went to check and it seems that we will only be able to upload the full revised manuscript after the interactive review stage. Should I copy and post some of the text I added from your previous comments here for you to have a look at? I could indicate the line numbers where the text has been added or are you happy to wait to see the final revised manuscript? I am happy to assist either way. Thank you.

---

## Referee Comment (RC3) · Anonymous Referee #1 · 8 Feb 2021

I am happy to wait for the final revised ms.

––––––––––––––––––––––––––––

---

## Referee Comment (RC4) · Anonymous Referee #2 · 26 Apr 2021

General Comments: The manuscript is nicely written with extensive reference to research articles and the authors clearly identifies 4 research questions which will be covered in the paper.

Scientific Comments: 1. However reading through the manuscript, missing the discussion over the scalability as shown in Figure 1c which would help about answering to the research question 4) What is the scalability of a rectangular grid, SWAN set-up?

2. Further as mentioned on Line 169 quote "The scalability is presented via three performance metrics: the efficiency, speed-up ratio and the timesaving ratio" would like that authors touch upon all the scalability for all these metrics and not only speed-up-ratio?

3. Further, in line 66, it is mentioned quote "Here we build on the case study of Genseberger & Donners using results produced in the present study for southern Africa.."
but to me it seems that the present study discussion is more and more comparing the
results of current study to study of Genseberger & Donners - to me as mentioned ear-
lier are of different domain. If this is not correct, please explain in Methodology and
Background accordingly.

4. To make this research article self standing – please include the case study domain
of southern Africa figure here (instead of refering to the - model configuration can be
found in (Rautenbach, et al., 2020 (a)) and (Rautenbach, et al., 2020 (b)). Also figure
of case study of Genseberger & Donners can be included here to make understanding
of the results and discussion clear to readers. Later , unless as pointed out in point 3
above.

5. Please include a table/figure in the Conclusion part to make conclusion more ob-
vious and readable to the users. Refer "A hybrid SWAN version for fast and efficient
practical wave modelling, Genseberger & Donners, (2020) paper section 4.2 to see
what I mean by including a table to compare between OpenMP / MPI different metrices
and/or with current study with the study of Genseberger & Donners.

6. Can authors make the connection between Zafari, Larsson, & Tillenius, (2019) study
of shallow water with the current study of SWAN Model clear. There is reference made
to "gcc" - but current study "Methodoloy and Background" does not include details of
this current study being run on gcc except what is mentioned in lines 124-128. The
reason for this comment is that seems that authors are hinting to gcc but no further
references or discussion on this in later sections. Maybe I am missing something here?

Technical Corrections:

Line 55: SLOSH : Sea, Lake, and Overland Surges from Hurricanes. (though SLOSH
can be NOAA official storm surge forecasting model - but this is not the official full
name) Line 58 : Mexican golf : I think here the Gulf of Mexico is being referred. Line
103: ration should be changed to ratio. Line 126: ggc should be changed to gcc. Line

156 : 16 nodes (16 × 25 threads) should be changed to 16 nodes (16 × 24 threads)
Line 157 : 64 nodes (16 × 24 threads) should be changed to 64 nodes (64 × 24 threads)

---

## Author Response (AR1)

Response: Thank you for your interest in our paper and for the discussion, which has helped to improve the manuscript. Herewith inline responses to the questions, comments, and corrections:

General Comments: The manuscript is nicely written with extensive reference to research articles and the authors clearly identifies 4 research questions which will be covered in the paper.

Scientific Comments: 1. However reading through the manuscript, missing the discussion over the scalability as shown in Figure 1c which would help about answering to the research question 4) What is the scalability of a rectangular grid, SWAN set-up?

Response: The scalability can be assessed in several ways. We use a variety of
metrics to help elucidate the concept. In general, the most dramatic increase in the time saving ratio occurs when the number of threads/ cores is below 6. Thereafter, increasing the number of cores doesn't substantially improve the time saving ratio. In general, the more thread/ cores one uses (for this SWAN configuration) the faster the total computation will be completed. This is evident from Fig. 2c (in the revised text). The point where the internal, domain-decomposed, communication time starts to dominate (and thus decrease run time with increase thread counts) is not reached within our thread counts (as explained on line 168). To be sure this is clear to the reader, a more detailed explanation has been added at the end of the discussion at line 176.

2. Further as mentioned on Line 169 quote "The scalability is presented via three performance metrics: the efficiency, speed-up ratio and the timesaving ratio" would like that authors touch upon all the scalability for all these metrics and not only speed-up ratio?

Response: Very good suggestion! This comment has been incorporated together with the previous comment at the end of the discussion section. Extra text has been added to discuss the performance in terms of these metrics.

3. Further, in line 66, it is mentioned quote "Here we build on the case study of Gense berger & Donners using results produced in the present study for southern Africa.." but to me it seems that the present study discussion is more and more comparing the results of current study to study of Genseberger & Donners - to me as mentioned earlier are of different domain. If this is not correct, please explain in Methodology and Background accordingly.

Response: We extend on the results of Genseberger & Donners rather than simply compare them. Specifically, we use different scalability/ benchmarking metrics than they did. We summarise their results because our studies build on their work (lines 66 -71). We make detailed comparisons between our study and theirs, to highlight the

new insights provided by our study.

4. To make this research article self standing – please include the case study domain of southern Africa figure here (instead of refering to the - model configuration can be found in (Rautenbach, et al., 2020 (a)) and (Rautenbach, et al., 2020 (b)). Also figure of case study of Genseberger & Donners can be included here to make understanding of the results and discussion clear to readers. Later , unless as pointed out in point 3 above.

Response: The South African SWAN model domain has been added as the new Figure 1. We could readily do this as this was our model. Due to copyright constraints I do not think its possible to add the model domain and extent of Genseberger and Donners. Due to the clear referencing, we recommend the readers to have a look at their publication, which is easily accessible.

5. Please include a table/figure in the Conclusion part to make conclusion more obvious and readable to the users. Refer "A hybrid SWAN version for fast and efficient practical wave modelling, Genseberger & Donners, (2020) paper section 4.2 to see what I mean by including a table to compare between OpenMP / MPI different metrices and/or with current study with the study of Genseberger & Donners.

Response: It was difficult to make a table that compares our results directly with Genseberger & Donners as they used multiple nodes and comparisons with their hybrid model. Our results were dependent on model resolution as well as number of threads/ cores (single node). We feel that a table could potentially confuse or obscure the key results. We note that Figure 1 (now Figure 2) now summarizes all the results, which, combined with the additional changes (added w.r.t the other comments above), hopefully improvers the readability of the conclusions.

6. Can authors make the connection between Zafari, Larsson, & Tillenius, (2019) study of shallow water with the current study of SWAN Model clear. There is reference made to "gcc" - but current study "Methodoloy and Background" does not include details of
this current study being run on gcc except what is mentioned in lines 124-128. The reason for this comment is that seems that authors are hinting to gcc but no further references or discussion on this in later sections. Maybe I am missing something here?

Response: This paper reference was added for completeness to inform the reader of other research that might be related to this topic. This paragraph aimed to introduce the next paragraph discussing the model resolutions used in the present study. A sentence was added to the conclusion suggesting further investigations with regards to using a gcc compiler.

Technical Corrections: Line 55: SLOSH : Sea, Lake, and Overland Surges from Hurricanes. (though SLOSH can be NOAA official storm surge forecasting model - but this is not the official full name) Corrected Line 58 : Mexican golf : I think here the Gulf of Mexico is being referred. Corrected Line 103: ration should be changed to ratio. Correction Line 126: ggc should be changed to gcc. Corrected Line 156 : 16 nodes ($16 \times 25$ threads) should be changed to 16 nodes ($16 \times 24$ threads) Line 157 : 64 nodes ($16 \times 24$ threads) should be changed to 64 nodes ($64 \times 24$ threads) Corrected

Geosci. Model Dev. Discuss.,
https://doi.org/10.5194/gmd-2020-314-SC2, 2021

[Figure]

Thank you for your interest in our paper and for the discussion. Herewith in-line responses to the questions, comments and corrections: SC1: Please provide the version number of SWAN in the title of your revised manuscript. As the websites cited in the articles code availability section are not persistent archives, please provide a persistent release for the exact source code version used for the publication in this paper. As explained in https://www.geoscientific-modeldevelopment.net/about/manuscript_types.html the preferred reference to this release is through the use of a DOI which then can be cited in the paper.

Author response: The version number has been added to the title. The current supported versions of SWAN are 41.20 and 41.31. From the following link we can see that the latest version of SWAN is fully compatible with version 40.91, used in the present

study (http://swanmodel.sourceforge.net/modifications/modifications.htm). The additions to the newer versions are just more choices in the parametrization models used to describe certain physical phenomena. Thus, executing the model with this latest version, together with the settings presented in this study, should produce identical results. To be sure readers of the paper can readily download all versions (Linux, Windows etc.) of this code, the current version of the software will be added to a university website associated with this paper. A link to this permanent cite will be added to the final version of the paper. We are currently in the process of establishing that online location.

[Figure]

Geosci. Model Dev. Discuss.,
https://doi.org/10.5194/gmd-2020-314-SC3, 2021

[Figure]

Author response: Thank you very much for the very useful and insightful comments. It is much appreciated. Please find the inline responses below:

1) The notion "thread" is a bit confusing for the MPI adepts, which should be "core". May be a combi "thread/core" would be a better wording.

Author response: All references to "thread" have been replaced with "thread/ core". 2) In general, within a single node (containing a number of threads/cores) OpenMP is more efficient than MPI. So, contrary to the study of Genseberger and Donner (2015, 2020), the results of your study contradict this general statement. Do you have an explanation for this? Perhaps, you may add some technical info concerning the used hardware with respect to this aspect (memory I/O, network, etc.)

[Figure]

Author response: OMP performs better at small thread numbers where MPI does better at larger thread numbers. This is illustrated in Figure 1 (a) and (b). This was clarified in the text on line 160 to 165. In Genseberger they note: "So, for this hardware the OpenMP version is twice as efficient as the MPI version." It must be related to differences in the hardware. Details regarding the memory and network has been added in Methodology Section. They also looked at the Wadden Sea while this is a southern African benchmarking study. 3) One of the possible reason why a perfect speed-up cannot be obtained (see Fig. 1) is the domain partition of the computational grid and also the wet/dry (or active/inactive) points. The employed partition is the stripwise one which is because of the underlying parallel technique, namely the wavefront method. See Genseberger and Donner (2015) and also Zijlema (2005). The stripwise partition might not be the most optimal one with respect to the speed-up. In this specific case, it leads to a maximum of 6 threads/cores without too much sacrificing parallel efficiency. It would be good to highlight this aspect. Added reference: M. Zijlema. Parallelization of a nearshore wind wave model for distributed memory architectures. In Parallel Computational Fluid Dynamics - Multidisciplinary applications, pages 207-214. Elsevier Science, 2005. Do you have active/non-active grid points in your model schematization? Can you comment on this? Technical corrections, etc. line 103: ration -> ratio line 108: compliers -> compilers line 156: (16 x 25 threads) -> (16 x 24 threads) line 157: (16 x 24 threads) -> (64 x 24 threads)

Author response: Thank you for this great recommendation. These details and references have been added on lines 170 onwards, at the end of the Discussion section. Yes, the African continent computational points are inactive. This details has also been added in the Discussion section. All these corrections were made, thank you.

line 234: please change version number; also suggested to add the Technical Manual of SWAN besides the User Manual, as it contains the details of both physics and numerics Author response: Corrections made, and references added.

―――――――――――――

[Figure]

[Figure]

Geosci. Model Dev. Discuss.,
https://doi.org/10.5194/gmd-2020-314-SC4, 2021

[Figure]

I went to check and it seems that we will only be able to upload the full revised manuscript after the interactive review stage. Should I copy and post some of the text I added from your previous comments here for you to have a look at? I could indicate the line numbers where the text has been added or are you happy to wait to see the final revised manuscript? I am happy to assist either way. Thank you.
* * *

---

## Author Response (AR2)

The follow text has been added to ensure clarity regarding the reproducibility of the present study. Please referee to line 80 to 85.

All computations were performed on Intel Xeon Gold E5-2670, 2.3GHz computational nodes. Twenty-eight threads/ cores each with 96 GB RAM were used with1 Gbyte/s inter-thread communication speed. Given that the present study was performed using a single computational node, inter-node communication speeds are not considered. Thus, given a computational node with similar processing speed, the present study should be reproducible. In general, these node specifications are reasonably standard and therefore the present study is an acceptable representation of the SWAN scalability parameters.